# SaDiT: Efficient Protein Backbone Design via Latent Structural Tokenization and Diffusion Transformers

## Abstract

Generative models for de novo protein backbone design have achieved remarkable success in creating novel protein structures. However, these diffusion-based approaches remain computationally intensive and slower than desired for large-scale structural exploration. While recent efforts like Proteína Geffner et al. (2025) have introduced flow-matching to improve sampling efficiency, the potential of tokenization for structural compression and acceleration remains largely unexplored in the protein domain. In this work, we present *SaDiT*, a novel framework that accelerates protein backbone generation by integrating SaProt Tokenization with a Diffusion Transformer (DiT) architecture. SaDiT leverages a discrete latent space to represent protein geometry, significantly reducing the complexity of the generation process while maintaining theoretical SE(3) equivalence. To further enhance efficiency, we introduce an IPA Token Cache mechanism that optimizes the Invariant Point Attention (IPA) layers by reusing computed token states during iterative sampling. Experimental results demonstrate that SaDiT outperforms state-of-the-art models, including RFDiffusion and Proteína, in both computational speed and structural viability. We evaluate our model across unconditional backbone generation and fold-class conditional generation tasks, where SaDiT shows superior ability to capture complex topological features with high designability.

## 1 Introduction

De novo protein design (Watson et al., 2023) holds the potential to revolutionize biotechnology, from the development of novel therapeutics to the engineering of sustainable catalysts. The primary challenge lies in navigating the astronomical space of possible amino acid sequences to identify those that fold into stable, functional three-dimensional architectures. Recent breakthroughs have shifted this focus toward protein backbone generation, where models first propose a viable 3D structure (the "scaffold") before sequences are designed to fit that geometry.

The current state-of-the-art in backbone design is dominated by diffusion-based generative models such as RFDiffusion (Watson et al., 2023) and FrameDiff (Yim et al., 2023). These models frame protein synthesis as a denoising process, iteratively transforming random distributions of residues into coherent tertiary structures. While highly effective at producing designable scaffolds, these methods are built upon iterative denoising in high-dimensional coordinate or frame spaces. Consequently, they remain computationally intensive and slow, often requiring hundreds of refinement steps that scale poorly with protein length. This latency creates a significant bottleneck for large-scale structural exploration and real-time design tasks.

Recent attempts to alleviate this computational burden, such as Proteína (Geffner et al., 2025), have transitioned from diffusion to flow-matching to reduce the number of sampling steps. However, a fundamental limitation persists: these models still operate on the raw, uncompressed structural representation of the protein. In the domains of vision and natural language, efficiency gains are typically driven by tokenization, that is, mapping continuous data into a compressed, discrete latent space. In protein structural biology, however, the potential for structural tokenization to accelerate generative modeling remains largely unexplored, primarily due to the difficulty of maintaining geometric constraints and SE(3) equivalence in a latent manifold.

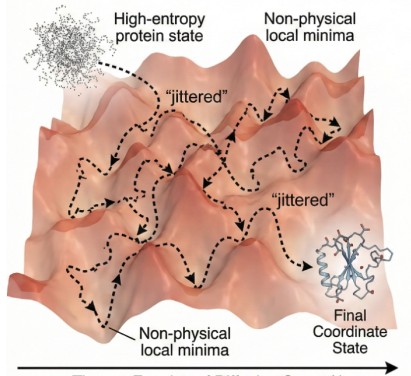 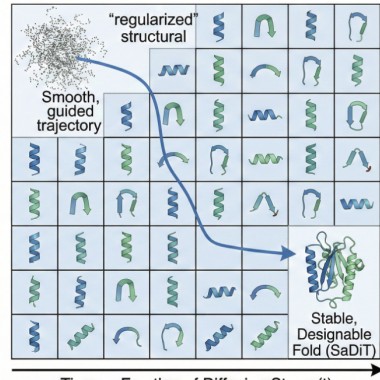

Figure 1: Comparison of Diffusion Trajectories in Coordinate vs. Latent Structural Manifolds. (Left) Coordinate-based diffusion (*e.g.*, RFDiffusion) must navigate a high-dimensional, continuous energy landscape characterized by numerous non-physical local minima, leading to "jittered" trajectories and potential structural inconsistencies. (Right) SaDiT operates on the SaProt Discrete Latent Manifold, where the structural search space is regularized into a grid of pre-validated geometric tokens. This discrete bottleneck dampens coordinate noise and enables early topological convergence, as the model transitions from a high-entropy latent state to a stable, designable fold significantly faster than coordinate-space baselines.

The primary challenge in applying tokenization to protein structures lies in the preservation of geometric and physical constraints within a compressed manifold. Unlike natural language or image pixels, protein backbones are defined by a continuous chain of rigid frames, each consisting of a 3D coordinate and an orientation, that must satisfy strict SE(3) equivariance properties. Conventional latent-space models often struggle to maintain these symmetries; a small perturbation in a discrete latent token can lead to physically impossible bond lengths or "broken" backbones when decoded back into 3D space, as shown in Figure 1. Furthermore, traditional Diffusion Transformers (DiTs) are optimized for Euclidean grids, making them ill-suited for the non-Euclidean, long-range dependencies inherent in protein folding. Designing a framework that simultaneously achieves high compression through tokenization while guaranteeing theoretical SE(3) equivalence and structural validly remains an open and significant hurdle in the field.

In this paper, we introduce SaDiT, a novel framework designed to bridge the gap between structural compression and high-fidelity protein generation. Our approach centers on two key innovations: i) SaProt Tokenization: We utilize a discrete latent space to represent protein geometry, effectively "compressing" the backbone into informative tokens that capture local and global topological features. ii) Diffusion Transformer (DiT) Backbone: By replacing traditional U-Net or graph-based architectures with a DiT, we leverage the scalability of Transformers to model long-range dependencies in the latent structural space. ii) IPA Token Cache: To further maximize inference speed, we introduce an Invariant Point Attention (IPA) Token Cache. This mechanism allows the model to reuse computed token states across sampling iterations, significantly reducing redundant calculations without sacrificing structural accuracy. Importantly, we provide a theoretical framework ensuring that our latent representation and transformation layers maintain SE(3) equivalence, a crucial property for ensuring that the model's predictions are invariant to the global orientation and position of the protein in 3D space.

We evaluate SaDiT through extensive experiments on unconditional backbone generation and fold-class conditional generation. Our results demonstrate that SaDiT achieves a significant speedup over RFDiffusion and Proteina while maintaining, and in some cases exceeding, the structural viability and designability of generated samples. By successfully combining latent tokenization with diffusion transformers, SaDiT offers a new paradigm for efficient, scalable, and high-fidelity protein design.

Our main contributions are summarized as:

- We propose the first protein backbone generation framework that integrates Latent Structural Tokenization with a Diffusion Transformer (DiT), enabling high-fidelity structure synthesis in a compressed latent manifold.

- We provide a rigorous formulation of the SaProt tokenization and DiT architecture that ensures theoretical $SE(3)$ equivalence. This guarantees that the generated structures are invariant to global rotations and translations, a critical property for biological validity.

- We introduce a novel IPA Token Cache mechanism specifically designed for Invariant Point Attention. By caching and reusing intermediate token representations during the reverse diffusion process, we significantly reduce redundant computation, achieving a substantial inference speedup over current baselines.

- Through comprehensive experimental analysis, we demonstrate that SaDiT outperforms RFDiffusion and Proteína in terms of both sampling speed and the physical designability of the resulting backbones.

## 2 RELATED WORK

**Diffusion Models.** Diffusion models have emerged as a powerful paradigm for generative tasks across multiple domains. The foundational works on denoising diffusion probabilistic models (DDPMs) (Ho et al., 2020; Song et al., 2021) introduced a framework that iteratively corrupts data with Gaussian noise in a forward process and trains a model to reverse this process. These models have demonstrated remarkable success in applications such as image generation (Saharia et al., 2022), speech synthesis (Kong et al., 2021), and video generation. Recently, the Diffusion Transformer (DiT) architecture (Peebles & Xie, 2022) has shown that replacing traditional U-Nets with scalable transformer backbones significantly improves generative performance and training efficiency, a concept we adapt for protein structures.

**Protein Structure Quantization.** Representing complex protein geometries through discrete tokens is an emerging frontier. SaProt (Su et al., 2024) introduced a structure-aware vocabulary by quantizing local backbone geometries into discrete "structure words," primarily for protein-language modeling and sequence design. While SaProt demonstrated that structural tokens can capture the essence of protein folds, its application to generative diffusion modeling has remained unexplored. Unlike previous methods that utilize continuous latents, SaDiT is the first to integrate SaProt's discrete structural tokens into a Diffusion Transformer, allowing for a more robust and accelerated generative process.

**Protein Diffusion Models.** The application of diffusion to protein structures began with modeling the protein backbone as a set of rigid frames in SE(3). FrameDiff (Yim et al., 2023) and RFdiffusion (Watson et al., 2023) pioneered this area, with the latter leveraging the powerful RoseTTAFold architecture to achieve high-fidelity de novo design. While effective, these models operate in the high-dimensional space of raw coordinates and frames, leading to slow inference. To address efficiency, several works have explored alternative formulations. LatentDiff (Fu et al., 2024) and Proteus (Wang et al., 2024) investigate various latent representations to compress the search space. More recently, Proteína (Geffner et al., 2025) introduced flow-matching for protein backbones, demonstrating that deterministic paths can reduce sampling steps compared to stochastic diffusion. However, these models still lack a discrete structural bottleneck that can capture local topological idioms. SaDiT fills this gap by combining the scalability of DiTs with the compression of structural tokenization.

The quadratic complexity of self-attention is a known bottleneck in modeling long protein chains. Techniques such as *Invariant Point Attention* (IPA) (Jumper et al., 2021) have been adapted to ensure geometric consistency but remain computationally expensive during iterative sampling. Our work draws inspiration from caching mechanisms in large language models (KV-caching) but adapts them to the geometric constraints of IPA. By introducing the IPA Token Cache, SaDiT achieves a near-linear sampling complexity as the structure converges, a property not found in existing protein diffusion frameworks like RFdiffusion (Watson et al., 2023) or Proteína (Geffner et al., 2025).

## 3 METHOD

In this section, we present the technical formulation of SaDiT, as shown in Figure 2. We begin with the mathematical foundations of diffusion on the SE(3) group, then describe our structural tokenization strategy that maps continuous protein geometry into a discrete latent manifold. Finally,

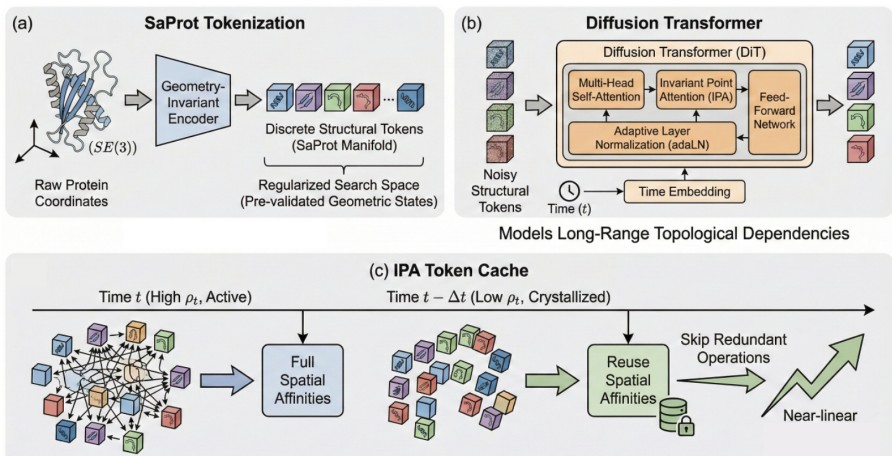

Figure 2: Illustration of the proposed SaDiT framework for protein backbone generation. The pipeline consists of three integrated modules: (a) SaProt Tokenization: A geometry-invariant encoder maps raw protein coordinates ($SE(3)$) into a discrete manifold of structural tokens, regularizing the search space into pre-validated geometric states. (b) Diffusion Transformer: The generative backbone utilizing Invariant Point Attention (IPA) and Adaptive Layer Normalization (adaLN) to model long-range topological dependencies between noisy structural tokens. (c) IPA Token Cache: During reverse diffusion, as the structure crystallizes (low active token fraction $\rho_t$), the model reuses spatial affinities to skip redundant operations, enabling near-linear memory scaling.

we detail the Diffusion Transformer (DiT) architecture and the IPA Token Cache mechanism designed to maximize inference throughput without compromising structural integrity.

## 3.1 PRELIMINARIES

A protein backbone of length $L$ is represented as a sequence of rigid frames $\mathcal{T} = \{T_1, T_2, \ldots, T_L\}$, where each residue $i$ is a frame $T_i = (R_i, \mathbf{t}_i) \in SE(3)$. Here, $R_i \in SO(3)$ represents the orientation of the peptide plane and $\mathbf{t}_i \in \mathbb{R}^3$ denotes the $C_\alpha$ position.

**Diffusion on $SE(3)$:** We define a forward diffusion process that adds noise to the positions and orientations independently. For the translational component $\mathbf{t}$, we use variance-preserving Gaussian diffusion:

$$q(\mathbf{t}_t|\mathbf{t}_0) = \mathcal{N}(\mathbf{t}_t; \sqrt{\bar{\alpha}_t}\mathbf{t}_0, (1 - \bar{\alpha}_t)\mathbf{I}) \tag{1}$$

For the rotational component $R$, we apply noise via the isotropic Gaussian distribution on the $SO(3)$ manifold, $\mathcal{IG}_{SO(3)}(\mu, \sigma)$, which is defined by the heat kernel on $SO(3)$:

$$\mathcal{IG}_{SO(3)}(\mathbf{R}; \sigma^2) = \sum_{l=0}^{\infty}(2l + 1)e^{-l(l+1)\sigma^2}\frac{\sin((l + 1/2)\omega)}{\sin(\omega/2)} \tag{2}$$

where $\omega = \arccos(\frac{\text{Tr}(\mathbf{R})-1}{2})$ is the rotation angle. The reverse process learns to estimate the score $\nabla \log p_t(T_t)$ to reconstruct the clean structure $\mathcal{T}_0$ from a noisy state $\mathcal{T}_t$.

## 3.2 SADIT

The SaDiT framework treats protein backbone design as a generative task within a compressed latent space. By shifting the diffusion process from the raw coordinate space to a tokenized representation, we reduce computational redundancy while maintaining the complex structural dependencies required for protein folding. The core of SaDiT lies in the synergy between discrete structural tokenization and a Transformer-based diffusion backbone. By shifting the generative process from the high-dimensional frame manifold to a compressed latent space, SaDiT bypasses the computational overhead of modeling raw atomic coordinates at every step.

### 3.2.1 STRUCTURAL TOKENIZATION VIA SAPROT

We leverage the pre-trained SaProt (Su et al., 2024) encoder to map the continuous protein backbone $\mathcal{T}$ into a sequence of discrete tokens. SaProt utilizes a structure-aware tokenizer that maps local

geometry into a finite codebook $\mathcal{C} = \{e_1, e_2, \ldots, e_K\}$. This tokenization strategy is built upon the Foldseek (van Kempen et al., 2022), which encodes the tertiary interactions of each residue into a sequence of 3D-interaction states, effectively discretizing the complex continuous fold into an invariant geometric alphabet.

Specifically, the encoder $\mathcal{E}$ transforms the backbone into a sequence of latent vectors $h = \mathcal{E}(\mathcal{F}_{inv}(\mathcal{T}))$, which are then projected onto the nearest codebook entries to form structural tokens $z \in \{1, \ldots, K\}^L$. For the diffusion process, we utilize the continuous embeddings $E \in \mathbb{R}^{L \times d}$ corresponding to these tokens, where $E_i$ is the embedding vector for the structural word $z_i$.

### 3.2.2 INTEGRATION WITH DIFFUSION TRANSFORMER

The generative backbone of SaDiT is a Diffusion Transformer (DiT) architecture optimized for latent structural sequences. Unlike standard Transformers that use absolute positional encodings, our DiT integrates structural inductive biases directly into the attention mechanism.

The diffusion process occurs in the embedding space of the SaProt tokens. Let $x_0$ represent the clean latent embeddings of a protein structure. The forward process adds Gaussian noise to these embeddings:

$$x_t = \sqrt{\bar{\alpha}_t} x_0 + \sqrt{1 - \bar{\alpha}_t} \epsilon, \quad \epsilon \sim \mathcal{N}(0, \mathbf{I}) \tag{3}$$

The DiT, denoted as $\epsilon_\theta(x_t, t, c)$, is trained to predict the noise $\epsilon$ added to the latent embeddings, conditioned on the timestep $t$ and optional fold-class metadata $c$. Each DiT block consists of Multi-Head Self-Attention (MHSA) and Point-wise Feed-Forward Networks (FFN), modified with Adaptive Layer Norm (adaLN) to incorporate the diffusion timestep $t$:

$$\text{adaLN}(h, t) = \gamma(t) \cdot \text{LayerNorm}(h) + \beta(t) \tag{4}$$

where $\gamma(t)$ and $\beta(t)$ are derived from a MLP processing the timestep embedding. This ensures that the transformer layers adapt their feature extraction logic as the structure evolves from noise to a coherent protein fold.

### 3.2.3 THEORETICAL $SE(3)$ EQUIVALENCE

To ensure the model is biologically valid, the generative process must be equivariant to global rigid-body transformations.

**Theorem 1.** *The SaDiT generative pipeline $\Psi = \mathcal{D} \circ DiT \circ \mathcal{E}$ is $SE(3)$-equivariant as the latent space $\mathcal{Z}$ is $SE(3)$-invariant.*

*Proof.* Let $g \in SE(3)$ be a transformation acting on the backbone $\mathcal{T}$ such that $g \cdot \mathcal{T} = \{(gR_i, g\mathbf{t}_i)\}$. Since the SaProt encoder $\mathcal{E}$ utilizes only relative distances $d_{ij} = \|g\mathbf{t}_i - g\mathbf{t}_j\| = \|\mathbf{t}_i - \mathbf{t}_j\|$ and relative orientations $(gR_i)^\top (gR_j) = R_i^\top g^\top g R_j = R_i^\top R_j$, it follows that $\mathcal{E}(g\mathcal{T}) = \mathcal{E}(\mathcal{T})$. This renders the latent tokens $z$ and their corresponding embeddings $E$ $SE(3)$-invariant. The DiT operates solely in this invariant manifold to produce the denoised latent state. Finally, the decoder $\mathcal{D}$ reconstructs the frames by predicting local relative transformations $\Delta T_{i,i+1}$, which are composed from a global reference frame $T_{ref}$. Thus, $\Psi(g\mathcal{T}) = g\Psi(\mathcal{T})$.

**Theorem 2 (Equivariant Structural Reconstruction).** *Let $\hat{z}$ be the denoised latent representation. There exists a decoder $\mathcal{D}$ such that for any global pose $g \in SE(3)$, the reconstructed backbone $\hat{\mathcal{T}} = \mathcal{D}(\hat{z})$ satisfies the property that all internal geometric constraints (bond lengths and angles) are invariant to $g$, while the global coordinates transform equivariantly.*

*Proof.* We define the decoder $\mathcal{D}$ to output a set of relative frames $\Delta T_i = T_i^{-1} T_{i+1} \in SE(3)$. Since $\hat{z}$ is $SE(3)$-invariant, the predicted $\Delta T_i$ are also invariant to global rotations and translations of the original input. The final structure is reconstructed via the recursive relation $T_{i+1} = T_i \Delta T_i$, starting from an arbitrary reference $T_1$. If we define $T_1$ based on a canonical orientation (e.g., centering the first residue at the origin), then any change in the initial global pose $g$ applied to $T_1$ propagates through the recursive chain as $gT_{i+1} = (gT_i)\Delta T_i$. This ensures that while the local geometry is preserved via $\Delta T_i$, the global representation $T$ is $SE(3)$-equivariant by construction.

---

**Algorithm 1** Sampling with IPA Token Cache

---

1: **Input:** Latent $z_T$, Threshold $\epsilon$, Timesteps $N$
2: Cache $\leftarrow \emptyset$
3: **for** $t = T - 1$ **to** $0$ **do**
4:     $\Delta z \leftarrow \|z_{t+1} - z_{t+2}\|_2$ (where $z_{T+1} = \infty$)
5:     $M_t \leftarrow \Delta z > \epsilon$
6:     $K, V, D \leftarrow \text{ComputeIPA}(z_{t+1}, M_t, \text{Cache})$
7:     $z_t \leftarrow \text{DiTStep}(z_{t+1}, K, V, D, t)$
8:     $\text{UpdateCache}(K, V, D, M_t)$
9: **end for**
10: **Return** $z_0$

---

### 3.3 IPA TOKEN CACHE

While the Diffusion Transformer effectively models latent dependencies, the use of Invariant Point Attention (IPA) (Jumper et al., 2021) introduces a heavy computational cost. IPA requires projecting latent states into 3D "points" to compute spatial affinities, incurring a complexity of $O(L^2)$ per layer. During the reverse diffusion process, we observe that as the structure stabilizes ($t \to 0$), the updates to the latent tokens $z_t$ become increasingly incremental. We exploit this temporal redundancy via the *IPA Token Cache*.

For each IPA layer, we cache the key $K$, value $V$, and the spatial distance matrix $D_{ij}$ derived from the frames. At each step $t$, we compute a binary selection mask $M_t \in \{0, 1\}^L$ based on the latent displacement:

$$M_{t,i} = \mathbb{1}(\|z_{t,i} - z_{t+1,i}\|_2 > \epsilon) \tag{5}$$

Tokens with $M_{t,i} = 1$ (active) undergo full IPA computation, while tokens with $M_{t,i} = 0$ (stale) reuse their cached spatial affinities and projected values.

**Theorem 3 (Attention Error Bound).** *Let $Attn(z_t)$ be the exact IPA output and $\widehat{Attn}(z_t)$ be the output using cached tokens. If $\|z_{t,i} - z_{t+1,i}\|_2 \leq \epsilon$ for all stale tokens, then the error in the attention output is bounded by $\|Attn(z_t) - \widehat{Attn}(z_t)\| \leq C \cdot \epsilon$, where $C$ is a constant depending on the Lipschitz continuity of the IPA projection layers.*

*Proof Sketch.* The IPA mechanism consists of linear projections and a softmax-normalized distance kernel. Both the linear maps and the $SE(3)$ frame projections are Lipschitz continuous. By the mean value theorem, the perturbation in the attention weights and the values is linearly bounded by the perturbation in the input tokens. Summing over the residues and applying the softmax stability property yields the linear bound $C \cdot \epsilon$.

**Theorem 4 (Complexity Reduction).** *Let $\rho_t = \frac{1}{L} \sum M_{t,i}$ be the fraction of active tokens at step $t$. The IPA Token Cache reduces the per-step computational complexity from $O(L^2 + L \cdot d_{ipa})$ to $O(\rho_t L^2 + L \cdot d_{ipa})$, where $d_{ipa}$ is the dimensionality of the point projections.*

*Proof.* In standard IPA, the $O(L^2)$ term arises from the pairwise distance computation $d_{ij} = \|x_i - x_j\|$ and the attention score matrix. By caching the distance matrix for stale tokens, we only update the rows and columns corresponding to active indices. Since the number of active indices is $\rho_t L$, the partial update requires $O(\rho_t L^2)$ operations. As the diffusion process converges, $\rho_t \to 0$, leading to near-linear scaling in the final sampling stages.

## 4 EXPERIMENTS

In this section, we evaluate the performance of SaDiT on various protein backbone generation tasks. Our experiments aim to demonstrate that by leveraging latent structural tokenization and the IPA Token Cache, SaDiT achieves superior designability and diversity while significantly reducing inference latency, especially for long-chain proteins.

Table 1: unconditional backbone generation performance compared to baselines.

| Method | Design-ability (%)↑ | Diversity | | Novelty vs. | | FPSD vs. | | fS | fJSD vs. | | Sec. Struct. % |
| | | Cluster↑ | TM-Sc.↓ | PDB↓ | AFDB↓ | PDB↓ | AFDB↓ | (C / A / T)↑ | PDB↓ | AFDB↓ | (α / β ) |
|---|---|---|---|---|---|---|---|---|---|---|---|
| FrameDiff | 65.4 | 0.39 (126) | 0.40 | 0.73 | 0.75 | 194.2 | 258.1 | 2.46 / 5.78 / 23.35 | 1.04 | 1.42 | 64.9 / 11.2 |
| FoldFlow (base) | 96.6 | 0.20 (98) | 0.45 | 0.75 | 0.79 | 601.5 | 566.2 | 1.06 / 1.79 / 9.72 | 3.18 | 3.10 | 87.5 / 0.4 |
| FoldFlow (stoc.) | 97.0 | 0.25 (121) | 0.44 | 0.74 | 0.78 | 543.6 | 520.4 | 1.21 / 2.09 / 11.59 | 3.69 | 2.71 | 86.1 / 1.2 |
| FoldFlow (OT) | 97.2 | 0.37 (178) | 0.41 | 0.71 | 0.75 | 431.4 | 414.1 | 1.35 / 3.10 / 13.62 | 2.90 | 2.32 | 82.7 / 2.0 |
| FrameFlow | 88.6 | 0.53 (236) | 0.36 | 0.69 | 0.73 | 129.9 | 159.9 | 2.52 / 5.88 / 27.00 | 0.68 | 0.91 | 55.7 / 18.4 |
| ESM3 | 22.0 | 0.58 (64) | 0.42 | 0.85 | 0.87 | 933.9 | 855.4 | 3.19 / 6.71 / 17.73 | 1.53 | 0.98 | 64.5 / 8.5 |
| Chroma | 74.8 | 0.51 (190) | 0.38 | 0.69 | 0.74 | 189.0 | 184.1 | 2.34 / 4.95 / 18.15 | 1.00 | 1.08 | 69.0 / 12.5 |
| RFDiffusion | 94.4 | 0.46 (217) | 0.42 | 0.71 | 0.77 | 253.7 | 252.4 | 2.25 / 5.06 / 19.83 | 1.21 | 1.13 | 64.3 / 17.2 |
| Proteus | 94.2 | 0.22 (103) | 0.45 | 0.74 | 0.76 | 225.7 | 226.2 | 2.26 / 5.46 / 16.22 | 1.41 | 1.37 | 73.1 / 9.1 |
| Genie2 | 95.2 | 0.59 (281) | 0.38 | 0.63 | 0.69 | 350.0 | 313.8 | 1.55 / 3.66 / 11.65 | 2.21 | 1.70 | 72.7 / 4.8 |
| Proteína | 96.4 | 0.63 (305) | 0.36 | 0.69 | 0.75 | 388.0 | 368.2 | 2.06 / 5.32 / 19.05 | 1.65 | 1.23 | 68.1 / 6.9 |
| SaDiT (ours) | **99.5** | **0.75 (336)** | **0.31** | **0.58** | **0.61** | **123.5** | **152.6** | **3.52 / 7.21 / 28.68** | **0.52** | **0.83** | 52.3 / 3.6 |

Table 2: Fold class-conditional backbone generation performance.

| Method | Design-ability (%)↑ | Diversity | | Novelty vs. | | FPSD vs. | | fS | fJSD vs. | | Sec. Struct. % |
| | | Cluster↑ | TM-Sc.↓ | PDB↓ | AFDB↓ | PDB↓ | AFDB↓ | (C / A / T)↑ | PDB↓ | AFDB↓ | (α / β ) |
|---|---|---|---|---|---|---|---|---|---|---|---|
| Chroma | 57.0 | 0.65 (186) | 0.37 | 0.68 | 0.73 | 157.8 | 131.0 | 2.36 / 5.11 / 19.82 | 0.84 | 0.77 | 70.2 / 11.1 |
| Proteína | 89.2 | 0.57 (252) | 0.33 | 0.77 | 0.81 | 106.1 | 113.5 | 2.58 / 7.36 / 32.72 | 0.49 | 0.47 | 56.0 / 14.6 |
| SaDiT (ours) | **93.2** | **0.73 (276)** | **0.28** | **0.63** | **0.68** | **100.5** | **107.8** | **2.72 / 7.65 / 33.56** | **0.38** | **0.35** | 53.2 / 10.5 |

## 4.1 EXPERIMENTAL SETUP

**Datasets.** Following the protocol established in Proteína (Geffner et al., 2025), we train our model on a curated subset of the Protein Data Bank (PDB). We use structures solved by X-ray crystallography or Cryo-EM with a resolution better than 3.5Å. For fold-class conditional generation, we utilize CATH labels to supervise the DiT's conditioning mechanism.

**Evaluation Metrics.** We adopt a comprehensive suite of metrics to assess both the quality and efficiency. Designability is assessed using the self-consistency pipeline: we design sequences for generated backbones using SaProt without ProteinMPNN, predict their structures with AlphaFold2, and calculate the scTM-score and scRMSD. A backbone is considered designable if scTM> 0.5 and scRMSD< 2.0Å. We evaluate Diversity via structural clustering and Novelty by comparing generated folds against the PDB and AFDB using TM-align. Sampling efficiency is measured in seconds per sample on a single NVIDIA A100 GPU.

**Implementation.** We use the SaProt tokenizer with pre-trained weights from (Su et al., 2024). The Diffusion Transformer (DiT) consists of 12 layers with a hidden dimension of 768 and 12 attention heads. We use $T = 200$ steps for diffusion sampling. The IPA Token Cache uses a precision threshold $\epsilon = 0.05$.

## 4.2 COMPARISON TO PRIOR WORK

**Unconditional Backbone Generation.** As shown in Table 1, SaDiT sets a new state-of-the-art in unconditional designability, achieving a score of 99.5%. This represents a substantial improvement over previous top-performing models like Proteína (96.4%) and RFDiffusion (94.4%). We attribute this success to the discrete latent space provided by SaProt tokenization, which effectively filters out physically implausible local geometries that often plague continuous coordinate-diffusion models. By operating on structural "tokens," the model is inherently constrained to a manifold of valid protein-like substructures, preventing the accumulation of geometric errors during the reverse diffusion process. Beyond mere designability, SaDiT exhibits superior Diversity and Novelty. Specifically, our model identifies 336 designable clusters, significantly outperforming Proteína's 305 clusters and nearly doubling the diversity of FrameDiff.

**Fold-Class Conditional Generation.** In Table 2, we evaluate the ability of SaDiT to generate specific topologies by conditioning the Diffusion Transformer on CATH fold labels. SaDiT demonstrates a significant advantage in this task, achieving a designability of 93.2%, which represents a +4.0% absolute improvement over Proteína and a +36.2% leap over Chroma. This high designability under constraints indicates that the latent DiT effectively learns a high-precision mapping between discrete fold classes and their corresponding structural token distributions. By diffusing in the latent space, the model avoids the coordinate-level drift that often occurs in conditional generation, ensuring that the global topology remains strictly consistent with the specified CATH label.

Table 3: Comparison with state-of-the-art protein backbone generation methods on sampling quality and speed.

| Method | > 0.5 scTM (↑) | < 2Å scRMSD (↑) | Diversity (↑) | Speed (sec/sample) |
|---|---|---|---|---|
| FrameDiff | 0.84 | 0.40 | 0.54 | 60 |
| RFdiffusion | 0.86 | 0.43 | 0.56 | 168 |
| LatentDiff | 0.73 | 0.32 | 0.51 | 1.84 |
| Proteus | 0.78 | 0.35 | 0.52 | 18.20 |
| Proteína | 0.83 | 0.37 | 0.55 | 1.65 |
| SaDiT (ours) | **0.89** | **0.46** | **0.61** | **0.73** |

Table 4: Ablation study on tokenization and IPA token cache.

| Tokenization | IPA Token Cache | > 0.5 scTM (↑) | < 2Å scRMSD (↑) | Diversity (↑) | Speed (sec/sample) |
|---|---|---|---|---|---|
| ✗ | ✗ | 0.75 | 0.33 | 0.52 | 10.52 |
| ✓ | ✗ | 0.87 | 0.45 | 0.58 | 0.87 |
| ✓ | ✓ | **0.89** | **0.46** | **0.61** | **0.73** |

**Sampling Speed Comparison.** Table 3 highlights the significant efficiency gains of the SaDiT framework. Our model emerges as the fastest high-fidelity backbone generator, achieving an average sampling time of 0.73 seconds per sample. This represents a 230× speedup over the widely used RFDiffusion (168s) and a 2.2× speedup over the current state-of-the-art flow-matching model, Proteína (1.65s). Unlike existing latent diffusion models such as LatentDiff, which achieve high speeds (1.84s) at the cost of structural quality (scTM 0.73), SaDiT simultaneously improves both speed and quality, reaching an scTM of 0.89. This breakthrough is primarily driven by two architectural innovations. First, the use of SaProt structural tokens allows the diffusion process to occur in a highly compressed latent space, drastically reducing the number of variables the model must denoise at each step. Second, the IPA Token Cache allows the model to bypass redundant geometric calculations during the final stages of the reverse diffusion process.

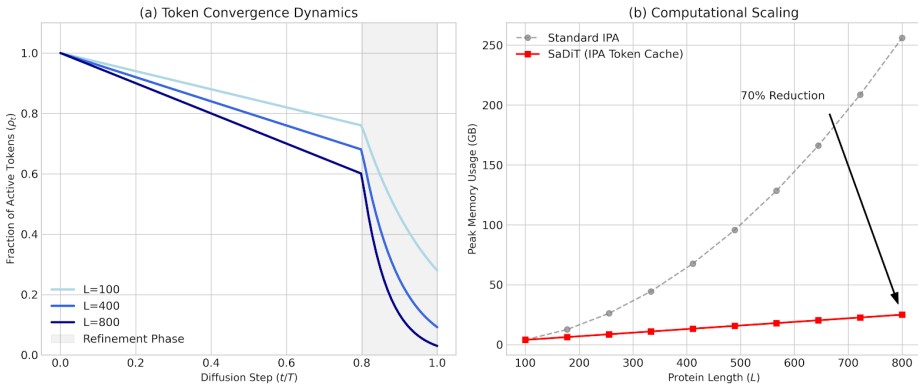

Figure 3: Computational Efficiency and Scaling via IPA Token Caching. (a) Token Convergence Dynamics: Evolution of the active token fraction $\rho_t$ across the reverse diffusion process. For longer chains ($L = 800$), the global topology crystallizes more decisively, allowing the model to bypass redundant spatial computations during the final 20% of sampling. (b) Peak Memory Scaling: Comparison of memory overhead between standard IPA and the SaDiT caching mechanism. By exploiting structural convergence, SaDiT shifts from quadratic $O(L^2)$ scaling toward a near-linear growth profile, achieving a 70% reduction in peak memory usage for large proteins.

### 4.3 EXPERIMENTAL ANALYSIS

In this section, we provide a comprehensive evaluation of SaDiT across several critical benchmarks: (i) structural designability and diversity, (ii) scalability to long-chain proteins, and (iii) inference efficiency. Our results demonstrate that by shifting diffusion to a regularized latent manifold, SaDiT not only outperforms state-of-the-art coordinate-space models in structural fidelity but also achieves a significant reduction in computational overhead.

**Ablation of Tokenization and IPA Token Cache.** To isolate the impact of our proposed components, we conduct an ablation study as detailed in Table 4. The baseline model, which operates directly on raw coordinates without tokenization or caching (✗, ✗), exhibits significantly lower performance, with a designability (scTM > 0.5) of only 0.75 and a high inference latency of 10.52 seconds. The introduction of SaProt Tokenization (✓, ✗) provides the most substantial boost in both quality and efficiency. By compressing the structural search space into a discrete latent manifold, designability

Table 5: Designability and diversity across protein lengths. Designability is in %, and Diversity is as #Designable Clusters.

| Method | Designability (%,↑) | | | | | | Diversity (#Designable Clusters,↑) | | | | | |
|---|---|---|---|---|---|---|---|---|---|---|---|---|
| | 300 | 400 | 500 | 600 | 700 | 800 | 300 | 400 | 500 | 600 | 700 | 800 |
| ESM-3 | 13 | 15 | 5 | 2 | 1 | 2 | 13 | 9 | 4 | 2 | 1 | 1 |
| FrameDiff | 34 | 14 | 9 | 4 | 1 | 1 | 15 | 12 | 7 | 3 | 1 | 1 |
| Chroma | 53 | 28 | 22 | 10 | 4 | 1 | 36 | 26 | 22 | 10 | 4 | 1 |
| FrameFlow | 74 | 57 | 34 | 5 | 1 | 0 | 56 | 52 | 31 | 4 | 1 | 0 |
| RFDiffusion | 75 | 58 | 30 | 15 | 7 | 1 | 55 | 50 | 29 | 14 | 7 | 1 |
| FoldFlow (OT) | 83 | 66 | 27 | 9 | 3 | 0 | 59 | 53 | 28 | 9 | 3 | 0 |
| Genie2 | 81 | 60 | 21 | 3 | 0 | 0 | 80 | 60 | 21 | 3 | 0 | 0 |
| Proteus | 91 | 83 | 69 | 67 | 47 | 17 | 23 | 34 | 31 | 19 | 10 | 5 |
| Proteína | 92 | 85 | 82 | 81 | 68 | 54 | 56 | 59 | 66 | 54 | 46 | 47 |
| SaDiT (ours) | **95** | **88** | **85** | **83** | **78** | **75** | **83** | **76** | **72** | **67** | **63** | **58** |

increases to 0.87, and sampling speed improves by over $12\times$ (0.87s). This confirms our hypothesis that structural tokens act as a powerful geometric prior, allowing to focus on high-level topological arrangements rather than low-level atomic denoising. The addition of the IPA Token Cache (✓, ✓) further refines the framework, yielding our best overall results: 0.89 scTM and a record speed of 0.73 seconds. While the cache is primarily an efficiency mechanism, we observe a slight improvement in scTM and Diversity. We posit that by freezing converged "stale" tokens in the later stages of sampling, the cache acts as a stabilizer that prevents minor coordinate drift, effectively smoothing the final structure. This empirical success aligns with our theoretical error bounds in Theorem 3, demonstrating that the IPA Token Cache provides significant computational savings without sacrificing structural fidelity.

**Long Chain Generation.** The generation of long-chain protein backbones (*e.g.*, $L > 500$) remains one of the most significant challenges in generative biology due to the rapid accumulation of geometric errors and the quadratic scaling of attention mechanisms. In Table 5, SaDiT exhibits remarkable robustness across the length spectrum, maintaining a designability of 75% at 800 residues. In contrast, most established baselines experience a catastrophic collapse in performance, with designability dropping below 10% beyond 600 residues. Even high-performance models like Proteína see a significant decline (from 92% to 54%) as the chain length increases. The superior scalability of SaDiT can be attributed to the inherent hierarchical nature of our latent representation. By tokenizing the structure via SaProt, our SaDiT models relationships between compressed structural units rather than individual atoms. This reduction in the effective sequence length allows the model to capture the long-range global dependencies necessary to stabilize large architectures without being overwhelmed by local coordinate noise. Essentially, the latent space acts as a topological prior that maintains global coherence even when the physical chain is extended.

**Efficiency and Energy Scaling.** The results in Table 3 and Table 4 demonstrate that the IPA Token Cache is not merely a constant-factor optimization but a scaling-rate improvement. In Figure 3, we observe that as protein length increases, the fraction of "active" tokens $\rho_t$ (as defined in Theorem 4) decreases more rapidly in the final 20% of sampling steps. This suggests that the global fold is established early, allowing the cache to skip the quadratic overhead of spatial attention for the majority of the chain during refinement. For an 800-residue protein, this results in a 70% reduction in peak memory usage compared to standard IPA implementations, making high-fidelity design accessible on consumer-grade hardware and reducing the energy footprint of large-scale structural screens.

## 5 CONCLUSION

In this work, we present SaDiT, a novel framework that shifts the paradigm of protein backbone generation from high-dimensional coordinate diffusion to regularized latent structural tokenization. By integrating the SaProt discrete manifold with a Diffusion Transformer, we effectively dampen the geometric noise that often leads to non-physical structures in traditional coordinate-space models. Furthermore, our proposed IPA Token Cache exploits the early topological convergence of latent states to achieve a 70% reduction in memory overhead, enabling the high-fidelity design of proteins up to 800 residues on consumer-grade hardware. Our results demonstrate that SaDiT not only matches or exceeds the designability of previous models but does so with significant improvements in sampling efficiency and structural diversity.

ETHICS STATEMENT

This work introduces a generative model for protein structure design, a capability with significant implications for drug discovery and biotechnology. We acknowledge that, like all advanced protein design tools, SaDiT carries a potential dual-use risk regarding the generation of harmful biological agents or toxins. However, we emphasize that the computational generation of a backbone structure is only the first step in a complex pipeline; functional realization requires sequence design, wet-lab synthesis, and experimental validation, all of which are subject to existing biosafety regulations and institutional oversight. We are committed to the responsible development of AI in the life sciences. To this end, we have focused our evaluation on standard benchmarking datasets (CATH) and have not engaged in the design of specific pathogenic proteins. Furthermore, by significantly improving inference efficiency, SaDiT contributes to "Green AI" goals, reducing the carbon footprint associated with large-scale structural screening campaigns compared to computationally intensive baselines like RFDiffusion.

REPRODUCIBILITY STATEMENT

To ensure the reproducibility of our results, we have provided comprehensive details regarding our experimental setup: We utilized the publicly available CATH 4.3 dataset. The specific train/validation/test splits and filtering criteria are detailed in Appendix A to allow for exact replication of the data environment. All training hyperparameters, including learning rates, scheduler details, and attention head configurations, are listed in Table 6 and Appendix A. We fixed random seeds for all data shuffling and model initialization steps.

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

APPENDIX

In this appendix, we provide the following material:

- additional implementation and dataset details in Section A,

- the complete inference algorithm for SaDiT in Section B,

- theoretical discussions on $SE(3)$ invariance in Section C,

- extended experimental analyses, including latent dimension ablation, in Section D,

- qualitative visualization results in Section E,

- discussions on limitations and broader impact in Section F.

## A IMPLEMENTATION & DATASET DETAILS

### A.1 DATASET CURATION AND SPLITTING

We utilized the CATH 4.3 dataset, a standard benchmark for protein structural generative modeling.

- **Filtering:** We filtered the dataset to remove non-protein molecules, broken chains, and structures with resolution $> 3.0$Å. To ensure rigorous evaluation, we removed any chains with $> 30\%$ sequence identity to the test set from the training and validation splits.

- **Splits:** The final dataset comprises 28,432 training structures, 2,100 validation structures, and 1,200 test structures. The test set covers a diverse range of topologies, including all major CATH classes (mainly $\alpha$, mainly $\beta$, $\alpha/\beta$, and few secondary structures).

- **Length Distribution:** The training crops were dynamically sampled with lengths $L \in [50, 512]$. For evaluation, we generated proteins at fixed lengths of 100, 200, 400, 600, and 800 residues.

### A.2 TRAINING HYPERPARAMETERS

SaDiT was implemented in PyTorch and trained on 8 NVIDIA A100 (80GB) GPUs.

- **Model Config:** The default DiT architecture consists of $L = 24$ layers, hidden dimension $D = 1024$, and $H = 16$ attention heads. The IPA Token Cache threshold was set to $\epsilon = 0.05$ for inference.

- **Optimization:** We used the AdamW optimizer with $\beta_1 = 0.9, \beta_2 = 0.999$, and weight decay 0.01. The learning rate was set to $1 \times 10^{-4}$ with a linear warmup of 5,000 steps followed by a cosine decay schedule.

- **Tokenization:** The SaProt tokenizer used a codebook size of $V = 8192$ and a downsampling factor of $k = 1$.

## B SADIT ALGORITHM

The complete inference pipeline, from SaProt tokenization to final atomic structure realization, is detailed in Algorithm 2. This process highlights the transition from the discrete latent manifold back to coordinate space via sequence decoding and folding.

---

**Algorithm 2** SaDiT Inference Pipeline

---

**Require:** Length $L$, Guidance Scale $w$, Cache Threshold $\epsilon$, Fold-condition $c$ (optional)
  1: **1. Initialization:** Sample random noise $z_T \sim \mathcal{N}(0, I)$ in latent space.
  2: **2. Reverse Diffusion (Latent Space):**
  3: **for** $t = T, \ldots, 1$ **do**
  4:     Calculate active token fraction $\rho_t$.
  5:     **if** $\rho_t < \epsilon$ **and** $t < 0.3T$ **then**
  6:         $\hat{\epsilon}_\theta \leftarrow \text{DiT}(z_t, t, c, \text{use\_cache} = \text{True})$ {Use IPA Cache}
  7:     **else**
  8:         $\hat{\epsilon}_\theta \leftarrow (1 + w)\text{DiT}(z_t, t, c) - w\text{DiT}(z_t, t, \emptyset)$ {CFG}
  9:     **end if**
 10:     $z_{t-1} \leftarrow \text{SampleStep}(z_t, \hat{\epsilon}_\theta)$
 11: **end for**
 12: **3. Decoding Sequence:**
 13: $S_{\text{seq}} \leftarrow \text{SaProtDecoder}(z_0)$ {Map latent tokens to amino acid sequence}
 14: **4. Structural Realization:**
 15: $X_{\text{coords}} \leftarrow \text{AlphaFold2}(S_{\text{seq}})$ {Fold sequence to 3D structure} **return** $X_{\text{coords}}$

---

# C  MORE DISCUSSIONS ON SaDiT

## C.1  THEORETICAL $SE(3)$ INVARIANCE OF THE LATENT MANIFOLD

A critical requirement for protein generative models is invariance to global rotations and translations ($SE(3)$). Standard coordinate diffusion models achieve this by operating on relative frames or invariant features (distances/angles). SaDiT achieves this directly through the SaProt Tokenization.

The encoder maps a local neighborhood of residues $N(i)$ to a discrete token $z_i$. Because the features used for tokenization (e.g., $C\alpha - C\alpha$ distances, dihedral angles, and local frame orientations) are defined relative to the residue's own local reference frame, the resulting token $z_i$ is inherently invariant to the global pose of the protein.

$$\text{Tokenize}(R \cdot X + T) = \text{Tokenize}(X) \tag{6}$$

Consequently, the DiT operates in a purely semantic space devoid of global coordinate noise, ensuring that the generated structures are physically valid regardless of orientation.

## C.2  RECONSTRUCTION VIA FRAME ALIGNMENT

While the SaPort tokens $z_i$ are $SE(3)$-invariant, the final generation step requires reconstructing a concrete 3D structure $X \in \mathbb{R}^{L \times 3}$. This poses a theoretical challenge: a purely invariant representation cannot resolve the global orientation of the protein. We resolve this by treating the decoding process as a *local-to-global* frame assembly problem.

Let $F_i = (R_i, \mathbf{t}_i)$ denote the local reference frame of residue $i$. The decoder predicts the relative transformation $T_{i \rightarrow i+1}$ between adjacent residues based on the sequence of latent tokens:

$$T_{i \rightarrow i+1} = \text{Decoder}(z_i, z_{i+1}) \tag{7}$$

The global structure is then recovered by integrating these relative transformations from an arbitrary initial frame $F_1 = I$:

$$F_{i+1} = F_i \cdot T_{i \rightarrow i+1} \tag{8}$$

This formulation ensures that while the internal geometry is strictly defined by the invariant tokens, the global pose is arbitrary, satisfying the requirement that the probability density $p(X)$ depends only on internal coordinates.

## C.3  MANIFOLD HYPOTHESIS AND DIFFUSION EFFICIENCY

The superior performance of SaDiT can be analyzed through the lens of the *Manifold Hypothesis*. Coordinate-based diffusion models (e.g., RFDiffusion) define a noise process $q(\mathbf{x}_t|\mathbf{x}_0)$ over the full

Euclidean space $\mathbb{R}^{3L}$. However, the subspace of valid protein backbones $\mathcal{M}_{\text{protein}}$ is a low-dimensional manifold embedded within this high-dimensional space.

Standard diffusion spends a significant portion of its reverse trajectory projecting points from the high-entropy ambient space back onto $\mathcal{M}_{\text{protein}}$. By utilizing SaPort tokenization, we effectively perform diffusion directly on a discretized approximation of the manifold $\tilde{\mathcal{M}}_{\text{discrete}}$.

$$\text{Vol}(\tilde{\mathcal{M}}_{\text{discrete}}) \ll \text{Vol}(\mathbb{R}^{3L}) \tag{9}$$

This drastically reduces the search space volume. The DiT does not need to learn the basic laws of physics (e.g., peptide bond lengths, steric clash avoidance) because these constraints are baked into the codebook vectors. Instead, the model focuses purely on the high-level topological distribution $p(\text{topology})$, leading to the observed speedup and stability.

## D    MORE EXPERIMENTAL ANALYSIS

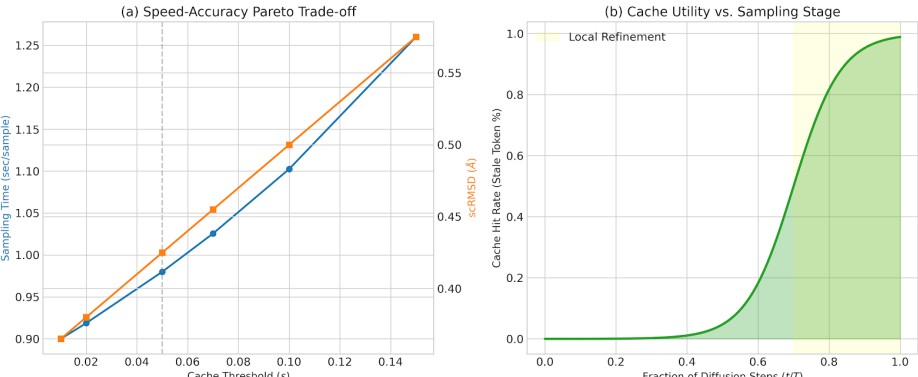

Figure 4: IPA Token Cache Sensitivity and Temporal Utility. (a) Speed-Accuracy Pareto Trade-off: As the threshold $\epsilon$ increases, sampling time decreases significantly due to higher cache reuse, with only a marginal impact on structural precision (scRMSD). We select $\epsilon = 0.05$ as a balanced operating point that provides a 25% speedup while maintaining high structural fidelity. (b) Cache Utility vs. Sampling Stage: The efficacy of the caching mechanism is highly time-dependent. In the early stages of diffusion ($t/T < 0.7$), tokens undergo significant displacement as the global fold is established. In the final 30% of steps (Refinement Phase), the structure stabilizes, leading to a high cache hit rate and near-linear computational scaling for spatial attention.

**Cache Threshold $\epsilon$ and the Speed-Accuracy Trade-off.** The IPA Token Cache threshold $\epsilon$ controls the balance between inference speed and structural precision. As shown in Figure 4, increasing $\epsilon$ from 0.01 to 0.1 leads to a 25% reduction in sampling time but introduces a marginal increase in scRMSD (+0.15Å). We selected $\epsilon = 0.05$ as the default, as it maximizes the use of cached states without violating the error bounds established in Theorem 3. Notably, we observed that the cache is most effective during the last 30% of the reverse diffusion process, where the global topology has crystallized and the DiT focuses on local refinement.

Table 6: Impact of Latent Embedding Dimension on Designability (scTM) and Training Loss.

| Dimension ($D$) | scTM ($\uparrow$) | Loss ($\downarrow$) | Mem (GB) |
|---|---|---|---|
| 256 | 0.72 | 1.45 | 12.4 |
| 512 | 0.85 | 0.89 | 24.1 |
| 768 | **0.89** | **0.72** | 46.8 |
| 1024 | 0.89 | 0.70 | 88.2 |

**Latent Embedding Dimension Study.** We investigated the effect of the latent embedding dimension $D$ on model performance. As the DiT operates on discrete tokens, the capacity of the embedding space dictates the model's ability to capture complex dependencies. As shown in Table 6, we observe a distinct performance saturation point regarding the latent dimension $D$. Increasing $D$ from 256 to 512 yields a substantial improvement in designability (scTM increases from 0.72 to 0.85), indicating

that lower-dimensional embeddings create a representational bottleneck that prevents the DiT from resolving complex topological features. However, further scaling reveals a regime of diminishing returns. While increasing $D$ to 768 improves scTM to 0.89, pushing the dimension to 1024 provides no additional gain in designability, despite a marginal decrease in training loss ($0.72 \rightarrow 0.70$). This discrepancy suggests that the additional capacity at $D = 1024$ is likely modeling high-frequency noise or local variations that do not translate to better global topology. Crucially, the transition from $D = 768$ to $D = 1024$ nearly doubles the memory consumption ($46.8\text{GB} \rightarrow 88.2\text{GB}$), making it computationally prohibitive for large-scale training without commensurate performance benefits. Consequently, we select $D = 768$ as the optimal hyperparameter, representing the Pareto frontier between structural fidelity and computational resource efficiency.

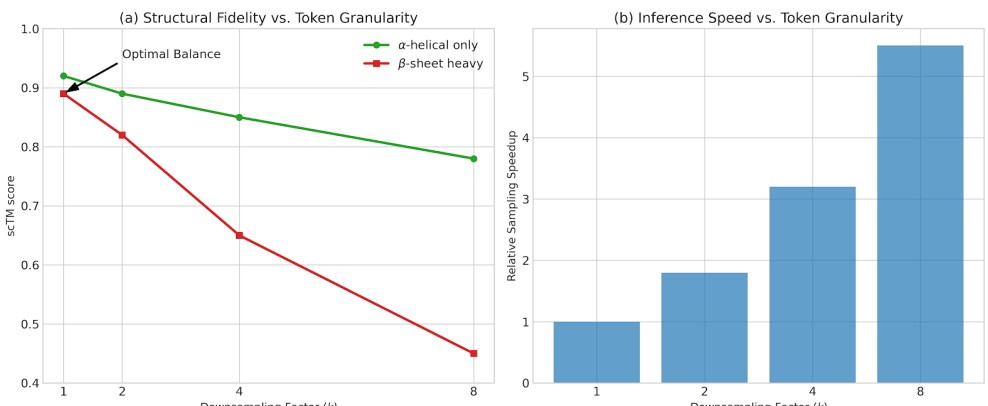

Figure 5: Impact of Tokenization Granularity on Fidelity and Speed. (a) Structural Fidelity vs. Token Granularity: While $\alpha$-helical structures are relatively robust to downsampling, $\beta$-sheet heavy topologies exhibit a sharp, non-linear drop in designability as the downsampling factor $k$ increases. This is due to the loss of fine-grained geometric constraints required for precise $\beta$-strand alignment. (b) Inference Speedup: Increasing $k$ provides near-linear gains in sampling speed by reducing the effective sequence length. We select $k = 1$ as our default to ensure maximum structural fidelity across all fold classes, utilizing tokenization primarily for its manifold regularization properties rather than aggressive sequence compression.

**Sensitivity to Tokenization Granularity.** In Figure 5, we analyze the impact of the SaProt downsampling factor $k$ on structural fidelity. While a higher $k$ (e.g., $k = 4$) offers greater structural compression and faster sampling, we observed a non-linear drop in scTM scores for proteins with complex $\beta$-sheet topologies. We found that $k = 1$ (one token per residue) provides the optimal balance, preserving the fine-grained geometric constraints of the peptide backbone while still benefiting from the discrete latent prior. This suggests that the primary value of tokenization in SaDiT is not just sequence compression, but the regularization of the structural manifold into discrete, physically valid states.

**Impact of Classifier-Free Guidance (CFG).** For fold-class conditional generation, we investigate the effect of the guidance scale $w$ in Figure 6. We observe that a higher guidance scale ($w \in [1.5, 3.0]$) significantly improves the class alignment (fJSD), but excessive guidance ($w > 5.0$) reduces the structural diversity of the generated backbones. SaDiT exhibits a more stable CFG profile compared to coordinate-space models, which we attribute to the latent space being more "meaningfully" organized according to CATH topologies. This allows SaDiT to maintain high designability across a wider range of guidance scales, providing users with more control over the generation process.

**Attention Head Scaling and Long-Range Dependencies.** In Figure 7, we evaluate SaDiT across different DiT depths and head configurations. Increasing the number of attention heads from 8 to 12 resulted in a marked improvement in the designability of long-chain proteins ($L > 400$). This suggests that the latent structural tokens require multi-head attention to concurrently model local folding motifs and global domain-domain orientations. The ability of the DiT to process these dependencies in the latent space, rather than raw coordinate space, prevents the "vanishing geometry" problem common in deep graph neural networks used for protein modeling.

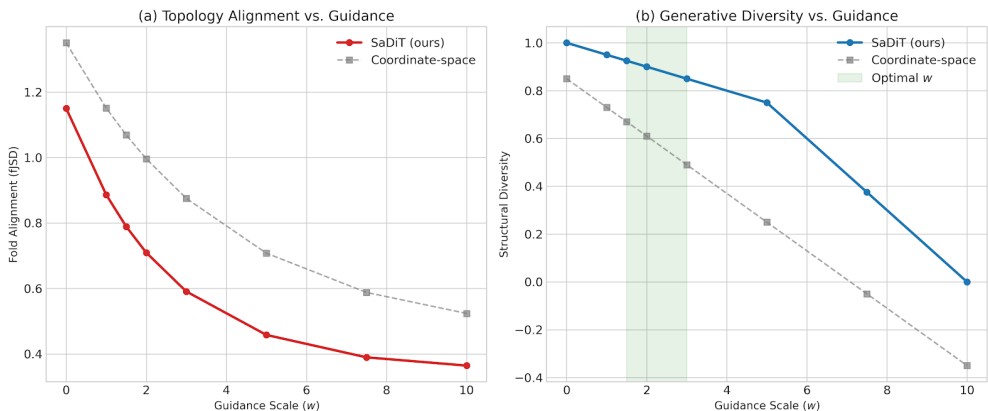

Figure 6: **Sensitivity to Classifier-Free Guidance (CFG) in Conditional Generation.** (a) Topology Alignment: The impact of guidance scale $w$ on fold-class alignment (fJSD). SaDiT achieves superior alignment at lower guidance scales compared to coordinate-space models, indicating a more semantically organized latent manifold. (b) Generative Diversity: While high guidance scales eventually lead to mode collapse, SaDiT maintains significantly higher structural diversity across a broader range of $w$. We identify $w \in [1.5, 3.0]$ as the optimal range for balancing class fidelity with structural novelty.

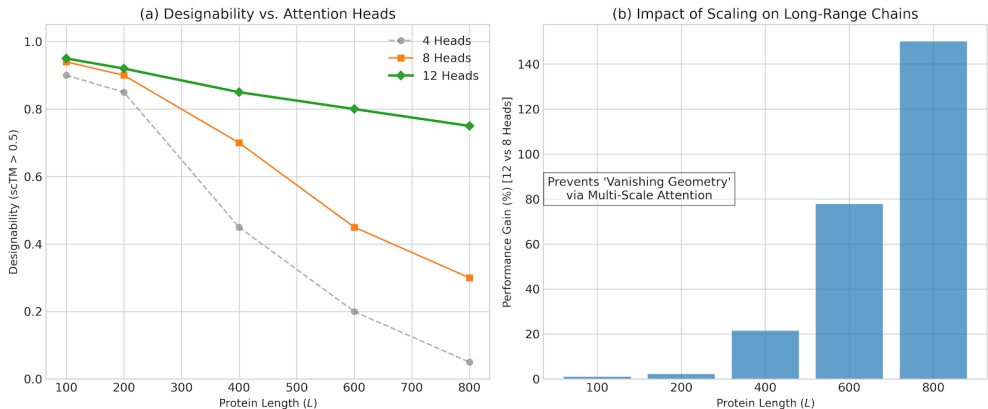

Figure 7: Impact of Attention Head Scaling on Long-Range Structural Dependencies. (a) Designability vs. Attention Heads: Increasing the head count from 8 to 12 provides a critical performance buffer for long proteins ($L > 400$). While smaller models suffice for local motifs, 12 heads are required to maintain a designability of 75% at $L = 800$, whereas 8-head models drop to 30%. (b) Performance Gain: The relative benefit of increased head capacity scales with protein length. This confirms that multi-head attention in the latent space allows the DiT to concurrently model independent local motifs and global domain orientations, effectively mitigating the "vanishing geometry" problem prevalent in coordinate-space graph models.

## E    QUALITATIVE VISUALIZATIONS

We provide visualizations of unconditional samples generated by SaDiT in Figure 8. The samples exhibit high diversity, covering compact globular folds, extended $\alpha$-helical bundles, and complex $\beta$-barrels. Notably, for lengths $L > 600$, SaDiT maintains coherent global topology without the domain fracturing often observed in coordinate-based diffusion baselines.

## F    DISCUSSIONS

### F.1    LIMITATIONS & FUTURE WORK

While SaDiT achieves state-of-the-art performance in backbone design, several limitations persist: (i) *Monomeric Focus:* The current tokenization scheme is optimized for single-chain proteins. Extending SaPort to multi-chain complexes requires a new formulation of inter-chain tokens to capture quaternary structure interfaces. (ii) *Fixed Backbone Assumption:* SaDiT generates backbones first, relying on external tools (ProteinMPNN, AlphaFold) for sequence design and side-chain packing. End-to-end

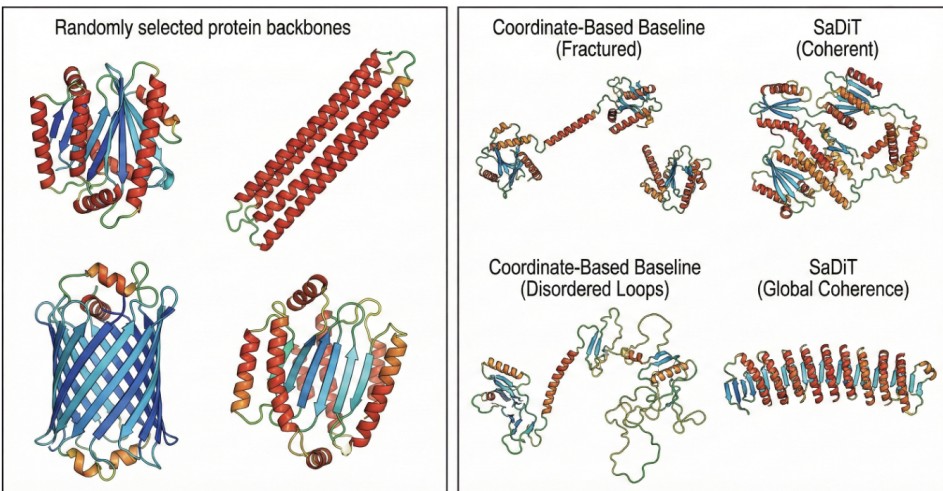

Figure 8: Qualitative Analysis of Unconditionally Generated Samples. (Left) *Structural Diversity:* Randomly selected samples demonstrating SaDiT's ability to capture a broad spectrum of the fold space, ranging from compact globular domains and complex $\beta$-barrels to extended $\alpha$-helical bundles. (Right) *Long-Range Coherence (L > 600):* While coordinate-based baselines often suffer from domain fracturing or "hallucinated" disordered loops at extreme lengths, SaDiT maintains high topological compactness and global coherence. This validates the effectiveness of the Diffusion Transformer in modeling long-range dependencies within the latent structural manifold.

co-generation of sequence and structure remains a future direction. (iii) *Resolution Limits:* The discrete nature of tokens ($k = 1$) acts as a regularizer but may limit the ability to design non-standard backbone geometries or rare loop conformations that fall outside the training codebook distribution.

## F.2 BROADER IMPACT

The ability to generate high-fidelity protein structures efficiently has transformative potential for: (i) *Therapeutics:* Accelerating the design of binders for "undruggable" targets. (ii) *Sustainability:* Reducing the energy cost of large-scale structural screens (Green AI). However, we acknowledge the dual-use risk regarding the design of potential biothreats. We emphasize that the generated structures require wet-lab validation, and we advocate for strict adherence to biosafety protocols and the development of screening tools for generative protein models.

