# OpenReview forum: "SaDiT: Efficient Protein Backbone Design via Latent Structural Tokenization and Diffusion Transformers"
_ICLR.cc/2026/Workshop/FM4Science — ICLR 2026 Workshop FM4Science Poster_

### Official Review · Reviewer_cEkm · 2026-02-16

**Rating:** 7
**Confidence:** 4

**Review:**

**Summary:**

This paper proposes SaDiT, a protein backbone generative framework that combines SaProt Tokenization with Diffusion Transformer (DiT) architecture, and uses an IPA Token Cache mechanism to accelerate SE(3)-aware attention. Compared to traditional diffusion-based protein generative models, SaDiT first maps protein backbones into a discrete structural token space, and then performs diffusion in embedding space. Experimental results show that the proposed method achieves strong unconditional and conditional generation performance on CATH, improves designability and reduces computational cost.

**Strengths:**

1. Compared to standard coordinate-based diffusion, the usage of SaProt tokenization and reconstructing via relative frames help the model incorporate more explicit geometric structure. The authors provide formal arguments regarding SE(3) equivariance and discuss complexity reductions introduced by the IPA Token Cache.

2. The experiments cover a range of settings. Multiple baseline models and metrics are used to show the proposed model's overall performance. The model's improvement is empirically significant. Authors also considered the efficiency, which is important for real scientific deployment.

**Weaknesses:**

1. While the method is technically and empirically strong, it is not entirely clear how SaDiT qualifies as a “scientific foundation model” rather than a task-specific generator. The model is trained and evaluated primarily on backbone generation within CATH. There is limited evidence of transferability across downstream tasks and different datasets. It'd be necessary and valuable to demonstrate a foundation model's generalization capability. Further analysis in this direction would strengthen the submission.

2. The experiments focus on in-distribution evaluation. Have you evaluated SaDiT under more challenging distribution shifts, such as unseen fold classes or rare topologies?

3. Because diffusion operates in a discrete structural token manifold, I wonder whether there is any measurable loss of geometric precision compared to continuous diffusion approaches. How does fine-grained backbone geometry compare to continuous coordinate-based diffusion models?

4. While backbone sampling is clearly faster, how significant is this in a full protein design pipeline, where AlphaFold or similar steps may be the dominant cost? Some discussion of pipeline-level runtime would help contextualize the practical impact.

---

### Official Review · Reviewer_qgCW · 2026-02-18
**SaDiT is a novel approach for protein backbone design.**

**Rating:** 8
**Confidence:** 4

**Review:**

- Summary
The authors describe SaDiT, a model that performs latent diffusion on structural tokens produced by an auxiliary structure eencoder (SaProt). The authors claim state-of-the-art performance across several critical sequence generation metrics (e.g. designability, diversity, novelty, etc.) and achieves improved inference sampling efficiency over existing methods.
- Contributions
1) IPA token caching for reduced redundant computation at later denoising timesteps.
2) Latent diffusion on structural tokens
3) Theoretical commentary displaying how SE(3) equivariance can be achieved in a discrete token space
- Pros
1) Inference time gains are markedly improved over existing methods.
2) Ablation study demonstrates that both structure tokenization (via SaProt) and the IPA token caching are critical for preserving sampling quality and reducing inference time sampling.
- Cons
1) Sequence quality metrics (e.g. 'Self-consistency pipeline') rely heavily on the SaProt structure encoder, which is also a component of SaDiT. Though the metrics indicate that SaDiT is superior across-the-board compared to other approaches, this may be a partially biased evaluation of generated backbone fitness given the methodology. This naturally works against methods such as RFDiffusion which produce far more than just a backbone.
- Decision
This work implements latent diffusion on structure tokens for protein backbone generation. While the evaluation methodology raises some questions, the work is novel and thorough. The paper considers several works adopting different approaches to the backbone generation problem and also ablates components of the introduced model for evaluating each part's necessity. Accept.

---

### Official Review · Reviewer_WHiJ · 2026-02-19
**Clear presentation, nifty contribution**

**Rating:** 9
**Confidence:** 4

**Review:**

The proposed model SaDiT accelerates de novo protein backbone generation by diffusing in a discrete, tokenized latent structural space (SaProt tokens) using a Diffusion Transformer, while preserving SE(3) equivariance. It adds an IPA Token Cache to reuse attention computations during sampling, reporting faster inference and strong designability vs prior backbone diffusion/flow models.
The paper proposes a clear efficiency story with concrete gains: the authors report ~0.73 sec/sample and large speedup vs RFDiffusion/Proteína while improving quality metrics (scTM/scRMSD, designability). Combining structural tokenization & diffusion transformer plus an IPA-specific caching mechanism is a coherent, differentiated contribution, and the unconditional + fold-class conditional results, ablations for tokenization/cache, and length-scaling up to 800 residues make for a throughout evaluation.

Some claims made in the Introduction feel however a bit overstated: “theoretical SE(3) equivalence” hinges on assumptions about encoder invariance + decoder construction; this statement specifically would benefit from tighter proofs/implementation details and empirical equivariance tests. Also, using SaProt for sequence design and AF2 for validation could bias toward tokenization-related priors, with baselines might not get equivalent downstream tooling.

---

### Meta-Review · Area_Chair_DSgQ · 2026-02-27

**Recommendation:** Accept (Oral)
**Confidence:** 4

**Metareview:**

Reviewers agreed that the contributions---latent diffusion with a Diffusion Transformer and IPA Token Caching---are both novel and valuable. Reviewers all agreed that the provided empirical evidence confirms these claims. A few suggestions to strengthen the evaluation were provided: examining and controlling for potential bias in backbone evaluation and evaluating generalization beyond in-distribution settings. The paper received unanimous accept decisions.

---

### Decision · Program_Chairs · 2026-03-03

Accept (Poster)